# Pathogenesis of Two Faces of DVT: New Identity of Venous Thromboembolism as Combined Micro-Macrothrombosis via Unifying Mechanism Based on “Two-Path Unifying Theory” of Hemostasis and “Two-Activation Theory of the Endothelium”

**DOI:** 10.3390/life12020220

**Published:** 2022-01-31

**Authors:** Jae C. Chang

**Affiliations:** Department of Medicine, Irvine School of Medicine, University of California, Irvine, CA 92868, USA; jaec@uci.edu; Tel.: +1-949-943-9988

**Keywords:** combined micro-macrothrombosis, deep vein thrombosis, endotheliopathy, hemostasis, immune thrombocytopenic purpura, macrothrombosis, venous thromboembolism, vascular microthrombotic disease

## Abstract

**Simple Summary:**

DVT is an intravascular blood clotting disorder that can be a life-threatening disease, particularly if it occurs in critically ill patients. Typically, distal DVT develops following a vascular injury associated with incidental trauma commonly involving lower extremities, which is transient and benign condition localized in the lower legs as solitary lesion. However, proximal/central DVT (i.e., venous thromboembolism) typically occurs in association with critical illnesses such as sepsis, diabetes, hypertension, cancer, autoimmune disease and others in the hospitalized patient, especially in the ICU. Recognition of different pathogenesis between distal DVT and proximal/central DVT is critically important because the prognosis is poorer in VTE. Its therapeutic approach should be different from distal DVT. The aim of this review is to identify the pathogenesis of two different types of DVT based on in vivo hemostatic mechanisms, which can explain their distinct phenotypes by clinical characteristics, laboratory data and imaging findings. An appropriate preventive measure can be put into the practice to avoid the onset of VTE. Additionally, should VTE be developed, proper and rational therapeutic regimen based on its pathogenesis can be designed for clinical trials to improve the outcome.

**Abstract:**

Venous thrombosis includes deep venous thrombosis (DVT), venous thromboembolism (VTE), venous microthrombosis and others. Still, the pathogenesis of each venous thrombosis is not clearly established. Currently, isolated distal DVT and multiple proximal/central DVT are considered to be the same macrothrombotic disease affecting the venous system but with varying degree of clinical expression related to its localization and severity. The genesis of two phenotypes of DVT differing in clinical features and prognostic outcome can be identified by their unique hemostatic mechanisms. Two recently proposed hemostatic theories in vivo have clearly defined the character between “microthrombi” and “macrothrombus” in the vascular system. Phenotypic expression of thrombosis depends upon two major variables: (1) depth of vascular wall damage and (2) extent of the injury affecting the vascular tree system. Vascular wall injury limited to endothelial cells (ECs) in sepsis produces “disseminated” microthrombi, but intravascular injury due to trauma extending from ECs to subendothelial tissue (SET) produces “local” macrothrombus. Pathogen-induced sepsis activates the complement system leading to generalized endotheliopathy, which releases ultra large von Willebrand factor (ULVWF) multimers from ECs and promotes ULVWF path of hemostasis. In the venous system, the activated ULVWF path initiates microthrombogenesis to form platelet-ULVWF complexes, which become “microthrombi strings” that produce venous endotheliopathy-associated vascular microthrombotic disease (vEA-VMTD) and immune thrombocytopenic purpura (ITP)-like syndrome. In the arterial system, endotheliopathy produces arterial EA-VMTD (aEA-VMTD) with “life-threatening” thrombotic thrombocytopenic purpura (TTP)-like syndrome. Typically, vEA-VMTD is “silent” unless complicated by additional local venous vascular injury. A local venous vessel trauma without sepsis produces localized macrothrombosis due to activated ULVWF and tissue factor (TF) paths from damaged ECs and SET, which causes distal DVT with good prognosis. However, if a septic patient with “silent” vEA-VMTD is complicated by additional vascular injury from in-hospital vascular accesses, “venous combined micro-macrothrombosis” may develop as VTE via the unifying mechanism of the “two-path unifying theory” of hemostasis. This paradigm shifting pathogenetic difference between distal DVT and proximal/central DVT calls for a reassessment of current therapeutic approaches.

## 1. Introduction

Venous thrombosis is typically recognized as “macrothrombosis” involving the veinous vessels of the circulatory system. Following a local intravenous vascular injury, it can initiate hemostasis and thrombosis in small, medium or large-sized veins, including venous sinuses as well as the pulmonary vasculature. Commonly, small, localized, isolated or distal, and deep or superficial vein macrothrombosis is transient and behaves as benign condition since it occurs as a result of normal hemostasis following an incidental vascular trauma that can be self-repaired utilizing normal physiological mechanism. Uncomplicated local thrombosis is resolved by natural fibrinolysis without clinical sequalae in the absence of underlying pathology such as sepsis, atherosclerosis, hypertension, autoimmune disease, diabetes or cancer [1,2].

In addition, there is another group of venous thrombosis developing in the microvasculature, including venules, venous sinusoids, and vasa vasorum, which is characterized by “microthrombosis” that occurs in sepsis or other critical illnesses in association of endotheliopathy-associated vascular microthrombotic disease (EA-VMTD) [3,4]. In this case, both arterial EA-VMTD (aEA-VMTD) and venous EA-VMTD (vEA-VMTD) may coexist when the underlying disease such as sepsis activates the complement components leading to the formation of membrane attack complex (i.e., C5b-9) that promotes arterial and venous endotheliopathy. Unlike aEA-VMTD that triggers life-threatening thrombotic thrombocytopenic purpura (TTP)-like syndrome [5], vEA-VMTD is “silent” and ‘transient” disorder because efficient physiologic dissolution of microthrombi takes place by ultra large usually large von Willebrand factor (ULVWF)-cleaving protease ADAMTS13 in slow venous circulation. Further, hypoxic organ dysfunction syndrome does not occur in venous microthrombosis. Even in significant septic endotheliopathy, commonly occurring vEA-VMTD may remain asymptomatic other than consumptive thrombocytopenia (i.e., ITP-like syndrome) unless it is complicated by additional vascular event resulting in complex arterial and/or venous thrombotic syndromes, which were sometimes encountered in SARS-CoV-2 sepsis [4,6,7,8,9,10].

During the COVID-19 pandemic, as predicted [11], the main clinical feature has been found to be acute respiratory distress syndrome (ARDS) characterized by microthrombosis in the pulmonary vasculature, but unexpectedly it was coexisted with “complex” form of macrothrombotic disorders such as pulmonary thromboembolism (PTE), and multiple large proximal/central catheter-associated deep venous thrombosis (DVT), venous thromboembolism (VTE), cerebral venous sinus thrombosis (CVST), inferior vena cava thrombosis (IVCT)/superior vena cava thrombosis (SVCT), Budd–Chiari syndrome, splanchnic vein thrombosis (SVT), and many others [6,7,8,9,10,12,13,14,15]. These complex thrombotic disorders have become a very important issue in the understanding of the character of each thrombotic disease as well as in managing of COVID-19 infection. No clear pathogenetic mechanisms have been identified yet, and medical community has been intrigued by the relationship of VTE and the character of ARDS [4,11]. The pathogenesis of inexplicable various thrombotic syndromes in COVID-19 infection has been a great challenge to this author, which led me to reexamine the mechanism of thrombogenesis producing the complex phenotypes of DVT by applying the hemostatic theories in vivo [4].

In this review, two different faces of DVT, distal DVT and proximal/central DVT (i.e., VTE), will be analyzed and their pathogenetic mechanisms be identified based on two hemostatic theories, “two-path unifying theory” of hemostasis and “two-activation theory of the endothelium” [16,17] with supporting laboratory data reported in the COVID-19 pandemic literature [4]. Proper clinical phenotype classification of thrombosis based on the pathogenesis and diagnostic characteristics will be presented, and therapeutic approach for prevention and potential management of VTE will be discussed.

## 2. Distal DVT vs. Proximal/Central DVT

The clinical features of DVT are influenced by the site of the vascular injury, depth of vascular wall damage, and extent of vascular tree involvement and affected organ as well as the size, shape, multiplicity and embolic manifestation of thrombosis. Depending upon distal vs. proximal/central location, small vs. large, simple vs. irregular, solitary vs. multiple, and non-embolic vs. embolic attributes, two major clinical phenotypes can be recognized as shown in Table 1 [18,19,20,21]. Tentatively, they will be grouped into (1) “distal DVT” for solitary/small thrombus typically involving infra-popliteal veins which is incidentally diagnosed without an underlying disease and (2) “proximal/central DVT” for multiple/large thrombi or long/extended thrombi involving one or more of central veins in the hospitalized patient with an underlying disease such as sepsis. Although their clinical severity and prognostic outcome are very different between distal DVT and proximal/central DVT, both phenotypes of DVT are conceptually accepted to be the same venous macrothrombosis initiated by tissue factor (TF)-FVIIa complex leading sequential activation of extrinsic coagulation cascade following an intravenous vascular injury. Nonetheless, unmistakable disparity in their clinical feature and prognosis has been the subject of controversial debate in regard to the therapeutic approach and perhaps contributed to the inconsistent result in interpreting of therapeutic clinical trials [22,23].

According to the hemostatic principle, DVT can be defined as thrombosis resulting from hemostasis in vascular injury in the venous system following the damage of ECs and subendothelial tissue (SET)/extravascular tissue (EVT), which genesis is explained in Three essentials in thrombosis and thrombogenesis shown in Table 2 derived from Three essentials of hemostasis [3,11]. Distal DVT usually occurs in the outpatient setting caused by isolated traumatic vascular injury resulting in activated extrinsic coagulation cascade leading to fibrin clot. On the other hand, proximal/central DVT (i.e., VTE) commonly occurs in the hospitalized patient and often characterized by large irregular thrombi at multiple sites. Although the hospitalization is known to be a major risk factor for serious DVT [24,25], especially in the intensive care unit setting [26,27,28], its pathogenetic difference from distal DVT has not been seriously evaluated from the point of view of molecular pathogenesis yet. Many determinants such as old age, immobilization, high altitude flying, drug, pregnancy, trauma and associated diseases such as diabetes, infection, autoimmune disease, cancer and sepsis have been thought to be risk factors. However, according to three essentials in thrombosis and thrombogenesis, the true risk “event” for thrombosis is only one, which is “vascular injury” due to a trauma. This injury is particularly common in the hospital resulting from vascular invasive “events”. Thus, the above listed determinants are not considered to be the risk factor for DVT, but are just contributing factors to vascular injury or potentiating factors for thrombosis only after vascular injury [17]. It is the hemostatic logic and natural law that thrombogenesis does not occur without ULVWF multimers and/or TF which are released into circulation following vascular wall damage. As illustrated in Figure 1 and Table 2, hemostasis cannot be initiated and thrombosis be produced without ULVWF and/or TF.

Multiple, large, and extended proximal/central DVT can be best defined as “VTE”. It usually occurs in the hospitalized patient and is much more serious DVT when compared to distal DVT. Hospitalization plays the key role for the development of VTE [29]. When a patient is admitted to the hospital, intensive care unit in particular, with critical illnesses such as sepsis, and cancer, in-hospital related vascular injury is very common event due to surgery, vascular access, procedure, vascular device, and mechanical ventilation therapy as well as in-hospital accident [4].

Local damage of the vessel wall would release two hemostatic components: ULVWF multimers from ECs and TF from SET/EVT. These two components initiate localized intravascular hemostasis; the former promotes microthrombogenesis by recruiting platelets via activated ULVWF path of hemostasis and assembles microthrombi strings, and the latter initiates fibrinogenesis by forming TF-FVIIa complex via activated TF path of hemostasis and produces thrombin, which converts fibrinogen to fibrin mesh/clot as depicted in Figure 1. These two sub-hemostatic paths finally merge together via unifying mechanism of macrothrombogenesis and produce venous macrothrombosis (e.g., distal DVT) in the venous system similar to arterial macrothrombosis (e.g., local arterial thrombosis) in arterial system [4,17]. However, Vascular injury occurring in the hospitalized patient with critical illnesses may take different course after in-hospital vascular access, which could lead to VTE distinctly different from distal DVT. To understand two different phenotypes of distal DVT and VTE, two seminal questions should be contemplated. First, why does serious VTE occur commonly in hospitalized patients? Second, what is the difference in the thrombus character between two different phenotypes of distal DVT and VTE? These questions can be answered by two hemostatic theories (Figure 1 and Figure 2) and Three essentials in thrombosis and thrombogenesis summarized in Table 2.

## 3. Understanding of Thrombosis Phenotypes

The term “thrombosis” has been used to describe “blood clot(s)” present within the circulatory system as a generic disease entity. According to contemporary dogma on hemostasis, every “thrombosis” in vivo is produced via extrinsic cascade progression of released TF from damaged SET/EVT following an intravascular injury which activates FVII to FVIIa and coagulation factors, including various serine proteases, protein substrates and fibrinogen. However, this simple concept of thrombogenesis based on TF activating extrinsic coagulation system alone cannot explain the clinical and pathological phenotypes of many different blood clots observed in clinical medicine and pathology. Thrombosis defined as the “blood clot” disorder includes the following diseases, but its inclusive term “thrombosis” is utterly inadequate to address clinical features and laboratory findings based on extrinsic TF pathway theory alone. The examples [18] are:Macrothrombosis developing in DVT, PTE, VTE, CVST, SVT, IVCT, SVCT and Budd–Chiari syndrome, portal vein thrombosis, acute ischemic stroke, acute myocardial infarction, aortic thrombus and renal vein thrombosis, etc.Microthrombosis developing in TTP, TTP-like syndrome, ARDS, diffuse encephalopathic stroke, hemolytic-uremic syndrome, transient ischemic attack (TIA), microaneurysm and thrombosis of the retinal artery, multiorgan dysfunction syndrome (MODS), hepatic veno-occlusive disease (VOD) and non-occlusive mesenteric ischemia (NOMI)Heparin-induced thrombocytopenia with “white clot syndrome” (HIT with WCS)Gangrene associated with arterial combined micro-macrothrombotic syndrome (e.g., symmetrical peripheral gangrene [SPG], purpura fulminans, Fournier’s gangrene in APL, Buerger’s disease, diabetic gangrene, Raynaud’s phenomenon, and acute necrotizing fasciitis, diabetic gangrene, etc.) [4,18]Fibrin clot disease occurring in acute promyelocytic leukemia (APL) and old term acute “disseminated intravascular coagulation (DIC)” occurring in EA-VMTD with hepatic coagulopathyHematoma and hemarthrosis within the tissue or cavitary areaConcurrent microthrombosis and macrothrombosis in both arterial and venous systems in paroxysmal nocturnal hemoglobinuria (PNH)

Certainly, many different forms of thrombosis do occur resulting from the complexity of hemostatic mechanisms affecting: (1) two sub-hemostatic paths (i.e., ULVWF and/or TF), (2) their unifying path to macrothrombogenesis, (3) venous or arterial systems, (4) microvasculature or larger vessel, (5) localization in the different vascular tree (i.e., localized and generalized, and solitary and multiple), and (6) altered hemostatic components associated with mutated genes (e.g., protein C, protein S, FV Leiden, and ADAMTS13, etc.). These combinations produce different characters of thrombi in the size, multiplicity, and involved tissue and organ as well as vascular system, and determine various clinical phenotypes [16,17] as summarized in Table 2. Therefore, all inclusively designated term “thrombosis” cannot provide the clinically explicative diagnosis for each thrombotic disorder because the term thrombosis only means an “existential” blood clotting disorder.

Based on two novel hemostatic theories shown in (Figure 1) [30] and (Figure 2) [31], variable pathologic and clinical phenotypes produced by thrombosis can be clearly and meaningfully identified in clinical practice. For example, although distal DVT and VTE are recognized as the same character disease composed of macrothrombus in the venous system, they are two different diseases expressing as distinct pathophysiological phenotypes (Table 1 and Table 3).

Now, DVT, how do two completely different clinical phenotypes distal DVT and VTE occur in spite of the same macrothrombosis? To comprehend this undefined pathogenesis, the two basic concepts of hemostasis should be understood; one is the genesis of three thrombosis mechanisms, which are (1) microthrombogenesis, (2) fibrinogenesis, and (3) macrothrombogenesis illustrated in Figure 1, and the other is the changes in vascular physiology, which includes (1) physiologic function of hemostatic components from vascular wall injury depicted in Figure 3, (2) the extent of involvement of vascular tree system in the microvasculature and larger vessel, and (3) the extent of inclusion of neutrophil extracellular traps (NETs) in the arterial system and venous system [3,18]. Current discussion will be primarily focused on the venous thrombosis model to identify the pathogenetic disparity between two phenotypes of DVT.

## 4. Physiological Mechanisms Involved in the Genesis of Thrombosis

Each phenotype in thrombosis, except for a few aberrant thrombotic disorders such as TTP, APL and HIT with WCS, is initiated by the vascular wall damage following an intravascular injury, which triggers thrombogenesis via normal hemostasis in vivo. Surprisingly, the same few thrombogenetic mechanisms affecting different organs and tissues as well as different vascular wall damage produce many distinctly different phenotypes of the human disease [16]. For examples, the mechanisms of thrombosis involving the vessels of the brain can cause TIA, acute ischemic stroke, thrombo-hemorrhagic stroke, diffuse encephalopathic stroke, hemorrhagic stroke, vascular dementia, neuropsychiatric disorder, or coma [17]. Additionally, different characters of thrombosis involving variable tissues and organs produce different phenotypes such diseases as myocardial infarction, septic coagulopathy, antiphospholipid antibody syndrome, acute necrotizing pancreatitis, rhabdomyolysis, TTP-like syndrome, venous circulatory congestion syndrome, veno-occlusive disease, peripheral gangrene [3,4], and, of course, two faces of DVT.

### 4.1. Vascular Wall Physiology in Venous Thrombosis

As in arterial thrombosis generated by normal hemostasis [30], the major phenotypes of DVT are also determined by two major factors: (1) depth of the vessel wall damage and (2) extent and location of the injury affecting the vascular tree system. The venous vascular SET damage from external force (local trauma) or venous ECs damage (endotheliopathy) due to disease such as sepsis, hypertension, cancer and diabetes can triggers intravenous hemostasis. The “localized” vascular injury releases local TF from SET and ULVWF from ECs which produces “distal DVT” in situ. However, “disseminated” endotheliopathy releases a large quantity of ULVWF multimers in the vascular tree system which attract platelets and assemble microthrombi strings, producing “vEA-VMTD”.

Physiologically, solitary distal DVT is self-limited disease at the localized site. Also, “disseminated” venous endotheliopathy is a relatively benign disease because the thrombosis “effect” in the venous system is very different from that in the arterial system due to low pressure flow and slower circulation as well as afferent direction, and there is no need of oxygen delivery to the organ and tissue. Thus, mild clinical profile is apparent in venous microthrombosis (i.e., vEA-VMTD) in particular. Microthrombosis of vEA-VMTD does not cause microangiopathic feature and organ hypoxia. Further, protease ADAMTS13 efficiently cleaves ULVWF multimers in lower shear stress circulation of the venous system. Therefore, vEA-VMTD may not be manifested as clinical venous disorder other than “consumptive thrombocytopenia”, which has been referred as “ITP” associated sepsis and other microthrombotic diseases [32,33,34]. This condition can be termed better as “ITP-like syndrome”. Indeed, dissimilar to aEA-VMTD which is characterized by TTP-like syndrome with microangiopathic hemolytic anemia (MAHA) and MODS in addition to thrombocytopenia [5], vEA-VMTD is a “silent” clinical microthrombotic phenotype, but can become a hemorrhagic phenotype if severe thrombocytopenia of the platelet count less than 10,000/μL is developed. This identity is an extremely important concept, as thrombocytopenia in ITP in sepsis is not due to immune destruction of the platelet, but is the result of consumption of the platelet in the process of microthrombogenesis in venous endotheliopathy illustrated in Figure 2 [18]. This mechanism will be explained more convincingly later.

However, vEA-VMTD (i.e., acute ITP or ITP-like syndrome) in sepsis can be a condition of “the calm before storm” that can very quickly be transformed to a life-threatening VTE. In sepsis (e.g., COVID-19), if the patient with “silent” vEA-VMTD with “microthrombi strings” composed of platelet-ULVWF multimer complexes undergoes a vascular access (e.g., surgery, vascular procedure/device, or mechanical ventilation in ICU) triggering vascular damage, it would release of TF from SET/EVT of damaged venous vasculature and activate the TF-FVIIa complex-induced coagulation cascade (i.e., fibrinogenesis), which produces numerous “fibrin meshes”. These two components, “microthrombi strings” from “silent” vEA-VMTD and “fibrin meshes” from in-hospital vascular injury would interact together via the unifying mechanism even though they are derived from two completely different vascular injuries [18]. The unified macrothrombi complex made of these binary components would turn into VTE characterized by multiple, large, irregular and/or extended venous thrombotic disorder with contribution of NETs at local or regional sites, and often travel to the afferent direction toward the heart and pulmonary artery causing multiple PTE [4].

### 4.2. Hemostasis Leading to Microthrombosis and Macrothrombosis

According to the blood vessel wall-based model of hemostasis (Figure 3) [11], the damaged vessel is the site of true hemostasis (coagulation) producing a hemostatic plug “external thrombus” in the external bodily injury, leading to cessation of hemorrhage [30]. It is also the site of hemostasis (thrombogenesis) initiating intravascular blood clot “thrombus” in the intravascular injury [16]. Its histologic structures are divided into the endothelium (i.e., ECs), tunica intima, tunica media and tunica externa, and released components (i.e., ULVWF and TF) from vascular wall damage contribute to molecular hemostasis.

The character of macrothrombus is clearly different from that of microthrombi with dissimilar hemostatic components, and each expresses distinct clinical and pathologic phenotypes. In essence, microthrombi are composed of platelet-ULVWF complexes resulting from disseminated endotheliopathy in the microvasculature, and macrothrombus is made of the combined product of microthrombi, fibrin meshes and NETs of various blood cells, components and molecules at the region of local vessel injury site. A local arterial or venous vascular wall injury (e.g., trauma) triggers localized release of ULVWF multimers from ECs and TF from SET into the intravascular space. ULVWF multimers become anchored to local endothelial membrane and recruit the platelet, and activate ULVWF path of hemostasis. Separately, TF activates FVII to initiate TF path of hemostasis. The former produces localized microthrombi strings composed of the platelet-ULVWF complex via microthrombogenesis and the latter produces fibrin meshes composed of the thrombin-fibrin complex via fibrinogenesis, in which the unifying process of microthrombi strings and fibrin meshes form a localized macrothrombus via macrothrombogenesis. This localized macrothrombosis in the venous system becomes distal DVT [16,18,30].

Pathogen toxins in septicemia (e.g., S protein of SARS-CoV-2 viremia) activates complement system in the host leading to formation of C5b-9 (i.e., membrane attack complex [MAC]) to neutralize the pathogen and kills infected cells [35]. MAC may cause the membrane pore formation and/or functional changes on the innocent ECs of the host [36] if endothelial protecting CD59 is underexpressed, and initiate disseminated endotheliopathy limited to ECs [4,5,16]. Endotheliopathy promotes release of ULVWF multimers and FVIII, and activates platelets. The released ULVWF multimers into the venous circulation would be cleaved by the protease ADAMTS13 in normal person. However, if mild to moderate deficiency of ADAMTS13 is present due to heterozygous mutation of the gene or relative insufficiency of the protease, it results in excessive accumulation of ULVWF multimers. The uncleaved multimers recruit platelets to form microthrombi strings in circulation, leading to vEA-VMTD [3,18]. These strings become microthrombi complexes in circulation and perhaps on the endothelial membrane. Even though microthrombi are formed and present in circulation, unlike arterial microvascular microthrombosis, venous microthrombosis in circulation does not cause MAHA and organ hypoxia (i.e., MODS), and also ADAMTS13 readily cleaves ULVWF multimers in the venous system. Thus, transient or circulating venous microthrombi may not be manifested as a clinical disorder (Table 4). Perhaps these are the reasons why vEA-VMTD may remain “silent” compared to aEA-VMTD in sepsis although platelet consumption is common as seen in acute ITP and ITP-like syndrome [32,33,37,38].

Accordingly, at least two characters of venous thrombosis occur; one is local trauma causing solitary and localized macrothrombotic DVT (i.e., distal DVT), and the other is venous endotheliopathy producing asymptomatic and “silent” but “transient” disseminated microthrombosis leading to consumptive thrombocytopenia via microthrombogenesis (i.e., ITP-like syndrome). Then, a next question is what is the pathogenesis of multiple, large and extended macrothrombotic DVT (i.e., VTE). According to the unifying mechanism of hemostasis in vivo, asymptomatic vEA-VMTD can be transformed to inexplicably “serious” VTE if a septic patient is complicated by additional in-hospital related vascular injury [4,18]. Surgery, vascular access, vascular device, and ventilator support with tracheal intubation as well as incidental in-hospital trauma can lead to vascular wall damage, which unlike distal DVT can trigger “venous combined micro-macrothrombosis” characterized by laboratory data showing activated ULVWF path with increased release of ULVWF and FVIII and activated TF path with markedly elevated D-dimer.

### 4.3. Dissimilarity of VTE from Distal DVT

Distal DVT is distinguishable from VTE by several clinical features (Table 1); first, it is usually diagnosed in the outpatient setting, but VTE often occurs and is diagnosed in the hospitalized and critically ill patients; second, it is small, solitary and localized disorder without inflammation, but VTE tends to be large and multiple lesions with local extension and commonly is associated with inflammation such as fever, chills and pain supporting the involvement of inflammatory pathway (Figure 2); third, it occurs typically without underlying disease, but VTE occurs with preexisting vascular disease such as septic endotheliopathy, hypertension, diabetes, polytrauma, autoimmune disease or cancer. More tangibly, in addition to clinical disparity VTE is associated with significantly measurable abnormalities in the hemostatic laboratory study, which are summarized in Table 3 and as follows.

Thrombocytopenia [27,39,40,41,42,43,44,45]Increased expression and activity of ULVWF/VWF [46,47,48,49,50]Increased FVIII activity [46,47,48,49]Often reduced ADAMTS13 activity [46,47,50]Increased D-dimer if VTE is present [20,50,51,52], but usually normal in distal DVT [20,51,52,53]Sometimes positive ANA [54], anti-dsDNA [55], anti-phospholipid antibodies (APLA) [56,57], and anti-platelet factor 4 antibodies (APF4A) antibodies supporting endothelial dysfunction and neo-autoantigen formation [18].Elevated inflammatory markers such as interleukin (IL)-6, IL-8, and C-reactive protein (CRP) [58,59,60]

The above laboratory findings are absolutely consistent with underlying disseminated endotheliopathy that leads to (1) activated ULVWF (microthrombotic) path of hemostasis, (2) activated inflammatory pathway [61] and (3) potentially activated autoimmune pathway due to complement activation. Thus, these findings present in complex forms of VTE infer that underlying vEA-VMTD must be present before the onset of VTE. Additionally, thrombocytopenia explicitly confirms its mechanism is consumption of platelets from the activated ULVWF path of hemostasis [11,31,61,62]. Both overexpression of ULVWF/VWF and increased activity of FVIII are the result of endothelial release from Weibel–Palade bodies [3,4,11].

Because distal DVT is a localized macrothrombosis from a local trauma, it is typically solitary and short-lived transient lesion and should not cause the above laboratory changes occurring in disseminated endotheliopathy. However, if “silent” vEA-VMTD (i.e., ITP-like syndrome in sepsis) due to venous endotheliopathy is complicated by additional vascular injury activating TF path and leads to significant “fibrin mesh” production, it would transform to a serious life-threatening VTE. Then, what could be the pathogenetic mechanism of VTE such as PTE, CVST [12,63], and rarely observed SVCT/IVCT [44,64], SVT [13] and Budd–Chiari syndrome [14,65] in bacterial and COVID-19 sepsis and other critical illnesses?

In Table 5, a novel hemostatic theory and intrinsic character-based classification of thrombosis identifies three types of DVT, which are (1) distal DVT of macrothrombosis, (2) vEA-VMTD of microthrombosis, and (3) VTE of combined micro-macrothrombosis. Since disseminated vEA-VMTD is “silent” without clinical evidence of DVT, only two forms of clinically significant DVT can be recognized, which are distal DVT and VTE. Symptomatic microthrombotic disease occurring in hepatic veno-occlusive disease and paroxysmal nocturnal hemoglobinuria will not be discussed here since they are incompletely understood forms of thrombosis among DVT at this time. Based on “two-path unifying theory” of hemostasis, VTE presenting with large and multiple macrothromboses should be termed as “venous combined micro-macrothrombosis” produced by unifying mechanism of “microthrombi” strings from underlying venous endotheliopathy and “fibrin-meshes” from another venous vascular event after hospitalization. The pathogenesis is postulated and illustrated in Figure 4 and Figure 5 [3,4].

### 4.4. Pathogenesis of Combined Micro-Macrothrombosis Syndromes

The normal unifying mechanism between microthrombi strings and fibrin meshes following vascular injury (e.g., aneurysm caused thrombosis and distal DVT) is a localized natural process of hemostasis [30,31]. However, the same unifying process of “microthrombi strings” from septic endotheliopathy and “fibrin meshes” from in-hospital vascular injury is a complex pathologic process leading to combined micro-macrothrombosis because it results in multiple macrothrombi due to disseminated microthrombi. The unifying mechanism of binary components from two different origins seems to be a far-fetched proposition, but this pathogenetic mechanism logically explains clinical and laboratory findings of both arterial and venous combined micro-macrothrombotic disorders They present with striking clinical features, and raise a serious dilemma in clinical management as well as diagnosis. Arterial combined micro-macrothrombosis produces multiple gangrene syndromes (e.g., SPG), but venous micro-macrothrombosis produces venous circulatory congestion syndrome (e.g., VTE).

In the arterial system, “fibrin meshes” formed from activated TF path at the arterial vascular damage site (e.g., in hospital arterial access/device) travel to the tissue via efferent circulation into the peripheral microvasculature of the digits where they encounter “microthrombi strings” that have already been formed in the arterioles and capillaries of the sepsis-associated aEA-VMTD. The unifying process of “microthrombi strings” and “fibrin meshes” would form numerous “minute combined micro-macrothrombi” within small arteries and cut off the blood supply to the efferent microvascular tree from the level of every small artery to the terminal digits, which result in complete saturation of the terminal arterial system with numerous minute thrombi. This leads to well-demarcated gangrene at approximately the same level of multiple digits causing total anoxia without collateral circulation [3,4]. Hemoglobin molecules would be trapped terminally to the efferent direction of the tissue and digit, and completely be denatured, leading to methemoglobin and inorganic iron that turn to iron sulfide, and become deposited into the surrounding tissue. They transform the tissue to gangrene and produce desiccated digits due to total anoxia. This gangrene syndrome (e.g., SPG) from arterial combined micro-macrothrombosis is quite a spectacle even in the intensive care unit. This conceptual proposition in arterial combined micro-macrothrombosis has been hypothesized from the concept of “two-path unifying theory” in previous publications [3,4,18].

Similarly, in the venous system, “silent” acute vEA-VMTD other than thrombocytopenia (i.e., ITP-like syndrome) in sepsis can be transformed to venous combined micro-macrothrombosis if additional in-hospital venous vascular injury releases fibrin meshes. The result of the unifying process would be less dramatic because of two reasons; first, venous circulation is afferent so that combined micro-macrothrombi within the venous system returning to the heart does not cause tissue and organ hypoxia/anoxia; however, multiple thrombo-emboli in the pulmonary artery could be the end result of venous micro-macrothrombi from VTE after passing through the right ventricle of the heart. Second, the unifying process of “fibrin meshes” and “microthrombi strings” in the venous system is slower due to lack of shear stress and low venous pressure. For these reasons, it does not produce minute macrothrombi, but instead forms extended multiple large-sized blood clots which are made of multiple long, connected, irregular stripe-shaped venous micro-macrothrombi complexes at the vicinity of the intravenous injury [18]. This mechanism produces VTE. The complex form of DVT is a serious life-threatening disease because it embolizes into the heart and lungs promoting multiple PTE.

Although their binary character of the micro-macrothrombosis is the same, the different physiologic nature between arterial and venous systems assembles completely different phenotypes of combined “micro-macrothrombosis” and affect the different part of the body expressing unique clinical phenotypes shown in Table 5 and Table 6.

Succinctly, the pathogenesis of combined “micro-macrothrombosis” occurring with gangrene syndrome in the arterial system [66,67,68,69,70,71] and with VTE and CVST in the venous system of the septic patient (e.g., COVID-19) results in the same binary complexes of micro-macrothrombi, but their clinical phenotypes clearly different due to disparate vascular hemodynamics and functional characteristics between two different vascular systems. The mechanism of fibrinogenesis from venous or arterial vascular injury interacts with that of microthrombogenesis from endotheliopathy, leading to combined micro-macrothrombosis of arterial gangrene and VTE. This unifying hypothesis has solved many mysterious human diseases. The conceptual framework of interaction between localized fibrin meshes and disseminated microthrombi has been proposed based on “two-path unifying theory” of hemostasis and well-documented laboratory findings of VTE as shown in Table 3 [3,4,30,31]. The clinical features, laboratory findings and pathogenetic differences are summarized and novel classification system of thrombosis is presented in Table 5, Table 6 and Table 7.

## 5. Additional Lessons Learned from COVID-19 Associated Thrombotic Syndromes

The insights from the clinical and laboratory changes in various thrombosis syndromes in COVID-19 pandemic and application of the essentials in hemostasis as well as two hemostatic theories—the “two-path unifying theory” of hemostasis and “two-activation theory of the endothelium”—have identified the pathogenetic mechanism of combined micro-macrothrombosis [4]. In the analytic process, the critical role of hospitalization/ICU admission in contributing to the complexity of thrombogenesis has been unveiled and also the character of acute ITP identified.

### 5.1. Role of In-Hospital and ICU Vascular Injury on the Pathogenesis of VTE

In contrast to distal DVT, VTE can be defined for every DVT characterized by venous combined micro-macrothrombosis. From the beginning of COVID-19 pandemic, the most intriguing question has been what is the pathogenetic mechanism of coexisting several thrombotic syndromes? These includes; (1) ARDS caused by disseminated microthrombosis in the pulmonary microvasculature, (2) other MODS caused by microvascular microthrombosis in various organs, (3) VTE characterized by multiple, large, irregular, and extended macrothrombi in proximal and central veins, (4) inexplicable well demarcated peripheral gangrene syndrome affecting multiple digits, and limbs [4]. Since VTE and peripheral gangrene syndrome typically occurs after hospitalization, it was surmised that two factors contribute to the pathogenesis of complexity of thrombotic syndromes. One is the underlying disease promoting microthrombosis, and the other is additional vascular injury-induced macrothrombosis after admission to the hospital/ICU. The obvious implication was in-hospital vascular access-induced injury together with endothelial injury of sepsis orchestrates different phenotypes of thrombosis [4].

Another interesting question has been why pathogen-based vaccine primarily causes CVST than VTE. CVST developed in the outpatient setting without COVID-19 infection following recombinant ChAdOx1-S and Ad26. CoV2.S vaccination [72,73], but the vaccines did cause complex forms of DVT such as VTE only in few instances. Interestingly, European Medicines Agency removed the cases of DVT from the reported “thrombosis” associated with ChAdOx1-S vaccine after detailed review of the original series. Perhaps DVT was interpreted as secondary thrombosis occurred after admission to the hospital [72]. When we look back the medical literature, vaccine-induced thrombocytopenia with thrombosis was mostly resulted in CVST but without direct implication to VTE. On the other hand, both CVST and VTE have occurred in COVID-19 sepsis and other diseases [12,63,74]. Further, the past medical literature suggested CVST was associated with head injury [74,75,76], and the consistent laboratory findings in both vaccine-induced CVST and sepsis-associated VTE were characterized by increased FVIII activity and overexpression of VWF [48,49] confirming of an underlying endotheliopathy [46,47,48,49,50]. This analysis suggests that unreported incidental head injury could have led to post-vaccine CVST due to underlying vaccine-induced “vEA-VMTD” or preexisting unidentified “silent” vEA-VMTD. VTE was much more common than CVST in the hospital/ICU admitted patient with sepsis [18].

This interpretation of the role of head injury may explain why vaccine-induced thrombocytopenia with thrombosis in non-hospitalized individuals had produced CVST than VTE. In patients with mild to modest ADAMTS13 deficiency, sepsis or vaccine may provoke endotheliopathy that activates ULVWF path of hemostasis resulting in consumptive thrombocytopenia [18]. Additional incidental head trauma could cause CVST, or in-hospital vascular injury could trigger VTE via activation of TF path due to SET damage. The potential role of partial ADAMTS13 deficiency should be evaluated in the patient with CVST or VTE [46,47,77,78,79]. This occurrence of venous combined micro-macrothrombosis also emphasizes how a vascular injury plays a critical role in the development of VTE as well as CVST.

An additional intriguing question in the COVID-19 pandemic has been why more severe secondary vascular and tissue injury have occurred in COVID-19 sepsis causing complex thrombosis syndromes than in other sepsis of the past? In retrospect, the major contributing factors for higher incidence of VTE in COVID-19 could have been due to (1) a hyped pandemic that forced hospitalization and multiple vascular accesses for the diagnosis, monitoring, and treatment, (2) aggressive respiratory support with mechanical ventilation with tracheal intubation leading to pulmonary vascular and tissue damage [80,81,82,83], and (3) an increased prothrombotic state due to reactive thrombocytosis secondary to hyperactivated pulmonary megakaryopoiesis associated with ARDS [84,85,86].

### 5.2. New Identity of Acute ITP as ITP-like Syndrome Triggered by Endotheliopathy

The cause and pathogenesis of idiopathic/immune thrombocytopenic purpura (ITP) have been unknown, but it has been suspected as an autoimmune disease caused by pathophysiological mechanism of antibody-mediated and/or T cell-mediated platelet destruction [87,88,89,90]. Severe ITP is recognized as a bleeding disorder, but a concern has been raised that ITP might also be a thrombophilic state because of its increased risk of thrombosis [33,91,92,93]. Indeed, the concurrence of unexplained thrombocytopenia (i.e., acute ITP) and VTE in the septic patient (e.g., COVID-19 sepsis) has been well known, and the mechanism of thrombocytopenia has been debated [4]. Further, recently adenovirus-vectored pathogen-based vaccine against COVID-19 has triggered ITP in association with a phenotype of venous thrombosis CVST [18,94,95]. This mysterious relationship in the triangle of (1) pathogen/vaccine, (2) acute ITP and (3) complex forms of DVT (e.g., VTE and CVST) has become an important topic in COVID-19 pandemic [4,96].

Overexpression of ULVWF/VWF/VWF antigen and increased FVIII activity are well established as the important markers for endotheliopathy promoting ULVWF path of hemostasis [3,11]. Further, decreased ADAMTS13, thrombophilic condition in ULVWF path, has been associated with ITP [97,98] and VTE [45,47,50]. However, MAHA and MODS occurring in TTP-like syndrome in association with the arterial microvasculature (i.e., aEA-VMTD) are absent in ITP and VTE. These differences suggest certain pathogen (e.g., SARS-CoV-2) could more likely cause endotheliopathy in the venous system (i.e., vEA-VMTD) as illustrated in Table 4. The findings of overexpressed ULVWF/VWF, increased FVIII activity and decreased ADAMTS13 activity support that both ITP and TTP-like syndrome are the result of the same pathogenetic process triggered by endotheliopathy (i.e., EA-VMTD) with different phenotype expression. The paucity of MAHA and MODS in ITP different from TTP-like syndrome can be interpreted that ITP is mild form of EA-VMTD, but TTP-like syndrome is severe form of EA-VMTD. How could this discrepancy be possible? Table 4 summarizes the different characteristics between arterial endotheliopathy leading to aEA-VMTD that causes TTP-like syndrome and venous endotheliopathy leading to vEA-VMTD that causes ITP-like syndrome [5,18]. MAHA and MODS are absent in venous endotheliopathy due to low venous pressure and lack of shear stress as well as afferent blood flow, but only triggering consumptive thrombocytopenia dissimilar to TTP-like syndrome. Thus, it can be called “ITP-like syndrome”.

If a septic patient with ITP-like syndrome and “microthrombi strings” in vEA-VMTD encounters a vascular damage due to vascular access in the hospital/ICU setting [99,100], TF released from SET/EVT would activate FVII, leading to TF-FVIIa complex-induced coagulation cascade, and form “fibrin meshes” at the vascular damage site. These two dynamic hemostatic complexes initiate the unifying mechanism of “microthrombi” of septic vEA-VMTD and “fibrin meshes” of venous vascular injury, leading to a life-threatening multiple “venous combined micro-macrothrombosis” (i.e., VTE) (Figure 4 and Figure 5). ITP-like syndrome commonly behaves as benign thrombotic disorder (i.e., “silent” vEA-VMTD) as predicted, but can turn to malignant thrombophilic/thrombotic disorder (i.e., VTE, including CVST) [33,91,92] if complicated by additional vascular injury. Therefore, ITP-like syndrome resulting from consumptive thrombocytopenia is a prothrombotic disorder contributing to VTE, but also becomes bleeding disorder if severe thrombocytopenia occurs.

Similar to acute ITP in acute venous endotheliopathy of sepsis (i.e., ITP-like syndrome), the prominent finding of vEA-VMTD is consumptive thrombocytopenia perhaps in continuing state of low grade microthrombogenesis and microthrombolysis, which could also explain certain cases of chronic ITP that is characterized by low grade endotheliopathy associated with chronic cardiovascular disease [101,102,103,104]. This endotheliopathy in some of chronic ITP and acute ITP (ITP-like syndrome) produces consumptive thrombocytopenia from activated ULVWF path, but this could be only the beginning stage of more serious venous combined micro-macrothrombosis. The platelet via consumptive thrombocytopenia creates a strange bridge between thrombosis and bleeding in the cause-effect relationship of hemostasis in vivo [3,4,5,16,18]. What an irony ITP/TTP-like syndrome is as vascular disorder each producing thrombosis and bleeding!

## 6. Consideration for the Diagnosis, Prevention and Treatment of Venous Combined Micro-Macrothrombosis

This author proposes a novel classification of venous thrombosis based on hemostatic theories as illustrated in Table 7. In most cases, the differential diagnosis between distal DVT due to local macrothrombosis and VTE due to venous combined micro-macrothrombosis can be readily established by their clinical features, imaging studies and laboratory changes as summarized in Table 3 and Table 6. The following laboratory test results should not only confirm the diagnosis, and but also support the pathogenetic mechanism of venous combined micro-macrothrombosis. 

ThrombocytopeniaIncreased activity of FVIIIOverexpression of ULVWF/VWF/VWF antigenMarkedly increased D-dimer levelDecreased ADAMTS13 activity

To prevent VTE in the hospitalized patient, it should be understood and emphasized that the hospitalization and admission to the intensive care unit are the most important contributing factor to the complexity of thrombosis, especially in the patient with sepsis and other critical illnesses, which pathogenesis leads to activated complement-induced endotheliopathy that promotes vEA-VMTD and eventually leading to venous combined micro-macrothrombosis. Therefore, in-hospital vascular event should be minimized for all the patients with the safe vascular access technic, special vascular care directive and guideline before, during and after surgical and vascular events. Thromboprophylaxis may be necessary as indicated with an anticoagulant, but theoretically thromboprophylaxis is not needed once the patient is discharged from the hospital without evidence of DVT. Continuing anticoagulation therapy may be hazardous due to potential bleeding complication.

Distal DVT may be treated with a shot-term oral anticoagulant [105,106]. The diagnosis of VTE can be readily established with if evidence of endothliopathy is present. The treatment should be decisive and comprehensive with proper management of the underlying VMTD. For the confirmed diagnosis of VTE with evidence of underlying vEA-VMTD, in addition to TF path inhibiting anticoagulant, clinical trials with additional antimicrothrombotic therapy such as recombinant ADAMTS13, N-acetyl cysteine, and intravenous immunoglobulin (IVIG) would be appropriate to prevent microthrombogenesis and/or correct underlying endotheliopathy producing microthrombosis [4], These combined therapeutic approaches eventually may prevent not only both venous and arterial combined micro-macrothrombosis, but also may reverse venous combined micro-macrothrombosis. The therapeutic effects can be monitored with the above laboratory parameters.

As far as the management approach for ITP-like syndrome (acute ITP) is concerned, preventive effort for the bleeding and thrombosis should be started with the patient education stressing the importance of avoiding trauma and injury, and limiting vascular access. Non-bleeding patient with chronic ITP does not need platelet transfusion if the platelet count is above 20,000/μL. Raising the platelet count might be more detrimental to the patient because it may upregulate microthrombogenesis and endotheliopathy. Instead, close surveillance is recommended to determine the trend of vascular health using the platelet count, FVIII activity, ULVWF/VWF antigen expression, and D-dimer level. In ITP-like syndrome and TTP-like syndrome, the platelet count of less than 10,000/μL may be an emergency trigger for short term prophylactic platelet transfusion [107] along with IVIG and/or steroid. Since platelet transfusion is contraindicated in consumptive thrombocytopenia associated with active endotheliopathy, it may contribute to diffuse encephalopathic stroke [17,108] and even diffuse microvascular myocardial ischemia [109].

## 7. Conclusions

Deep venous thrombosis is a very common thrombotic disorder with two major clinical phenotypes. One is distal DVT, which occurs with an incidental vascular trauma leading to isolated local macrothrombosis. It is a benign condition and self-limited disease with excellent prognosis and requires only a short-term anticoagulation treatment. The other is VTE characterized by venous combined micro-macrothrombosis associated with an underlying vEA-VMTD in the patient with sepsis or other critical illnesses. It occurs as a result of additional venous vascular injury in the hospital setting, especially the intensive care unit, and is a life-threatening condition that requires carefully planned management program for its prevention and coordinated therapeutic approach including combined anitmicrothrombotic treatment and anticoagulant therapy. An urgent task force is needed for creating well-designed prevention and therapeutic trials based on the in vivo hemostatic concept to prevent venous combined micro-macrothrombosis and to reduce morbidity and mortality from VTE.

## Figures and Tables

**Figure 1 life-12-00220-f001:**
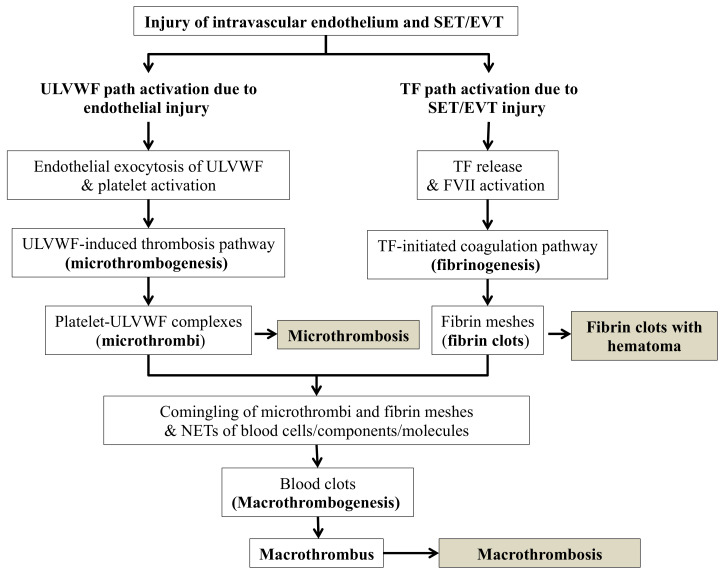
Normal hemostasis based on “two-path unifying theory” (updated and reproduced with permission from Walters Kluwer Health, Inc., and Chang JC: *Blood Coagul Fibrinolysis* 2018; 29:573-584). The concept of this theory is derived from physiological changes of vascular wall damage in vascular model of in vivo hemostasis based on three hemostatic fundamentals [11]. Nature has endowed human with only “one” normal hemostatic system to be available to abrogate hemorrhage. Hemostasis protects humans from accidental bleeding in external bodily injury and provides physiologic wound healing. Additionally, it warns human to avoid unnecessary self-inflicted injury and hostile environmental insults which cause intravascular injury that can lead to deadly thrombotic disorder. This is a true irony of nature—the same hemostatic mechanism not only provides life-saving wound healing, but also can lead to life-threatening thrombosis. In normal hemostasis, there must be two sub-hemostatic paths (ULVWF path and TF path), which have to be unified to make normal physiologic hemostasis to produce macrothrombus. In ECs damage due to disease such as sepsis, only ULVWF path is activated and leads to microthrombogenesis as seen in disseminated endotheliopathy (e.g., EA-VMTD), but, in ECs and SET damage occurring in a local trauma, both ULVWF path and TF path are activated locally and lead to fibrinogenesis and eventually leading to macrothrombogenesis to stop external bleeding or to form a localized distal DVT from internal bleeding. The microthrombogenesis produces microthrombi, which are incomplete blood clots but cause a very serious disease VMTD if disseminated. The localized macrothrombosis from trauma becomes localized blood clot, which becomes the hemostatic patch to stop the bleeding, and causes macrothrombus to lead to localized distal DVT. Abbreviations: DVT, deep venous thrombosis; EA-VMTD, endotheliopathy-associated vascular microthrombotic disease; ECs, endothelial cells; EVT, extravascular tissue; SET, subendothelial tissue; NETs, neutrophil extracellular traps; TF, tissue factor; ULVWF, ultra large von Willebrand factor multimers.

**Figure 2 life-12-00220-f002:**
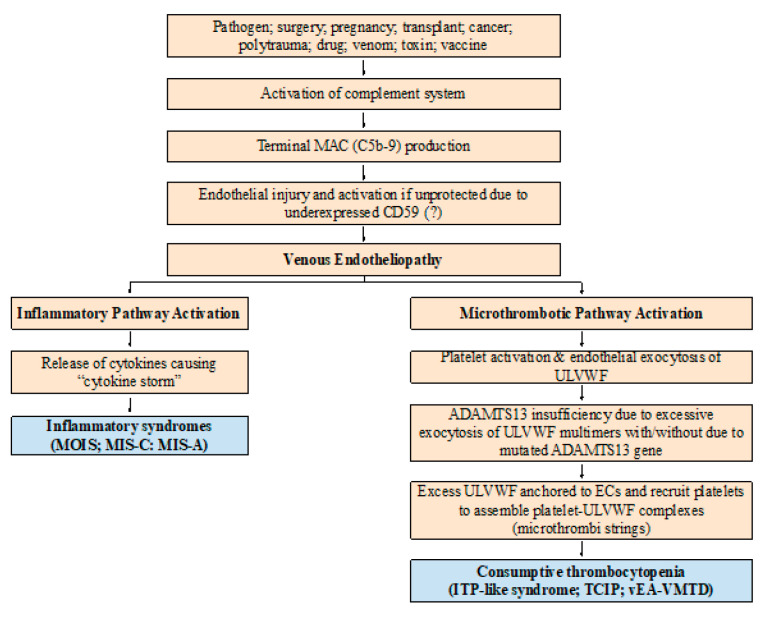
Pathogenesis of vEA-VMTD. The pathogenesis of endotheliopathy is established based on the “two-activation theory of the endothelium” and is self-explanatory. In the venous system, the molecular responses triggering exocytosis of ULVWF multimers and FVIII and release of cytokines in endotheliopathy are expected to be the same to arterial system. However, the physiological effects between venous and arterial circulation are very different in their shear stress force, vascular directional flow and role of oxygen delivery as noted in the text. Thus, the clinical phenotype of aEA-VMTD is typically TTP-like syndrome and MODS with consumptive thrombocytopenia, which is clinically serious, but that of vEA-VMTD is “silent” ITP with consumptive thrombocytopenia. In severe sepsis, TTP-like syndrome implies sepsis is complicated by overt “arterial” microthrombosis”, but in mild sepsis, ITP-like syndrome suggests transient “venous” microthrombosis has had occurred and perhaps is continuously being resolved by ADAMTS13. This difference has a very important implication in the understanding of combined micro-macrothrombotic syndromes between gangrene-producing arterial combined micro-macrothrombosis and VTE-producing venous combined micro-macrothrombosis. Abbreviations: vEA-VMTD, venous endotheliopathy associated vascular microthrombotic disease; aEA-VMTD, arterial EA-VMTD; ITP, immune thrombocytopenic purpura; MAC, membrane attack complex; MIS-A, multisystem inflammatory syndrome in adults; MIS-C, multisystem inflammatory syndrome in children; MOIS, multiorgan inflammatory syndrome; TCIP, thrombocytopenia in critically ill patients; TTP, thrombotic thrombocytopenic purpura; ULVWF, ultra large von Willebrand factor.

**Figure 3 life-12-00220-f003:**
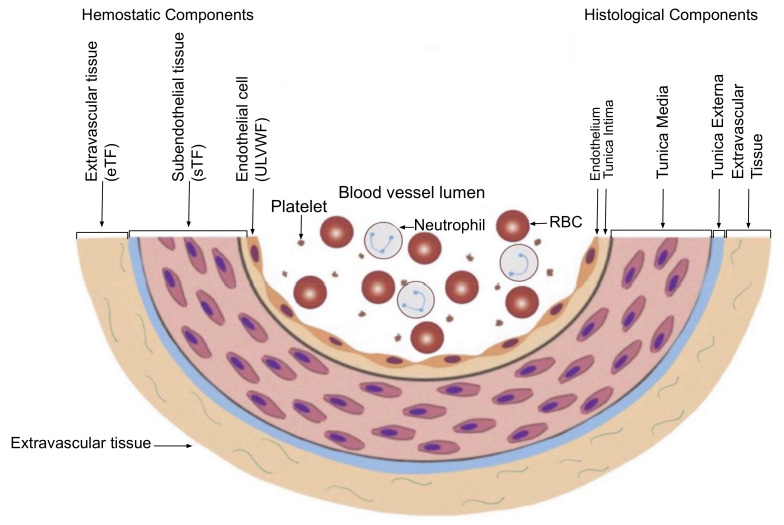
Schematic illustration of cross section of blood vessel histology and hemostatic components (Reproduced and modified with permission from Chang JC. *Clin Appl Thromb Hemost* 2019 Jan–Dec; 25:1076029619887437) [11]. The blood vessel wall is the site of hemostasis (coagulation) to produce blood clots (hemostatic plug) and to stop hemorrhage in external bodily injury. It is also the site of hemostasis (thrombogenesis) to produce intravascular blood clot (thrombus) in intravascular injury that causes thrombosis. Its histologic components are divided into the endothelium, tunica intima, tunica media and tunica externa, and each component has its function contributing to molecular hemostasis. As illustrated, ECs damage triggers exocytosis of ULVWF and SET damage promotes the release of sTF from tunica intima, tunica media and tunica externa. EVT damage releases of eTF from the outside of blood vessel wall. This depth of blood vessel injury contributes to the genesis of different thrombotic disorders such as microthrombosis, macrothrombosis, fibrin clots/hematoma and thrombo-hemorrhagic clots. This concept is especially important in the understanding of different phenotypes of stroke and heart attack. Abbreviations: EVT, extravascular tissue; eTF, extravascular tissue factor; SET, subendothelial tissue; sTF, subendothelial tissue factor; RBC, red blood cells; ULVWF, ultra large von Willebrand factor.

**Figure 4 life-12-00220-f004:**
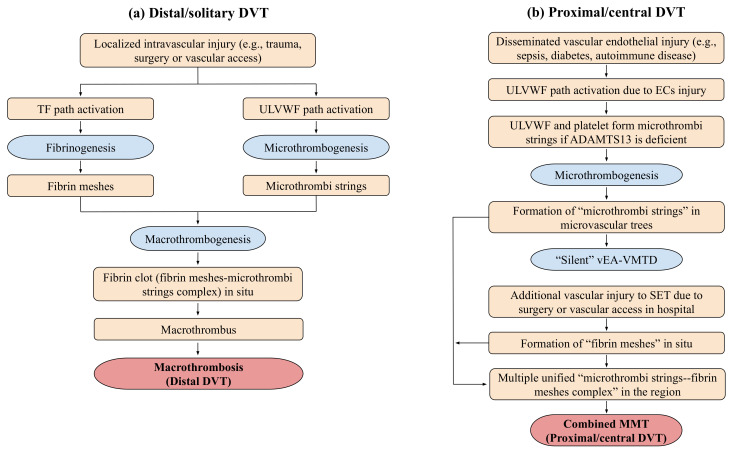
Genesis of two different DVT based on two-path unifying theory of hemostasis in vivo. Contemporary concept for every DVT is macrothrombosis in the venous system composed of blood clots containing fibrin clots, platelets, and NETs as a result of activation of TF-FVIIa complex-initiated coagulation cascade following vascular injury in the venous system. However, the exact mechanism producing macrothrombosis has not been determined, nor is the composition of macrothrombus proven. The “two-path unifying theory” of hemostasis and “two-activation theory of the endothelium” have been able to provide the exact role of major hemostatic components, including ULVWF, platelets, coagulation factors, and various cellular and molecular traps including NETs, in the vascular system. Based on hemostatic theories, two different phenotypes of DVT can be identified by their different pathogenesis and character of thrombosis. The genesis of distal DVT in Figure 4 (**a**) and that of proximal/central DVT (i.e., VTE) in Figure 4 (**b**) are summarized and elaborated in the text. Abbreviations: DVT, deep venous thrombosis; ECs, endothelial cells; MMT, micro-macrothrombosis; NETS, neutrophil extracellular traps; SET, subendothelial tissue; TF, tissue factor; ULVWF, ultra large von Willebrand factor; vEA-VMTD, venous endotheliopathy-associated vascular microthrombotic disease.

**Figure 5 life-12-00220-f005:**
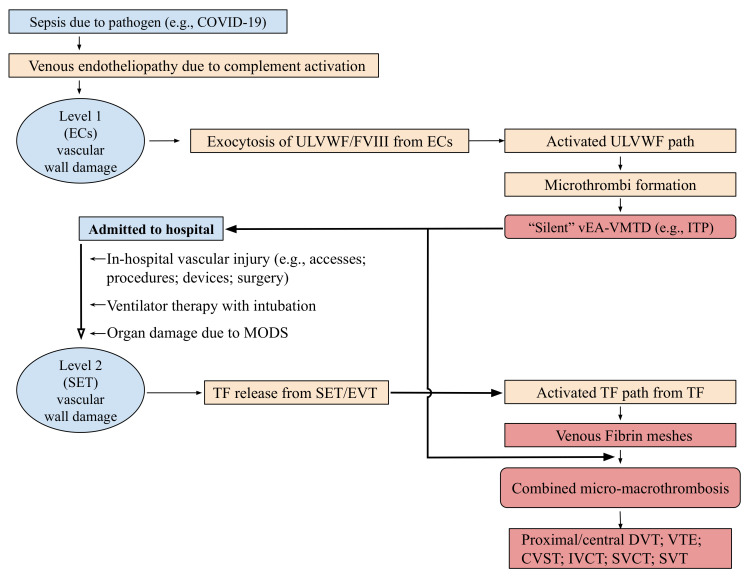
Pathogenesis of venous combined micro-macrothrombosis (Reproduced and modified with permission from Chang JC. *Vasc Health Risk Manag*. 2021, 17, 273–298) [4]. The Figure is self-explanatory. The pathogenesis of venous combined micro-macrothrombotic syndrome is the same as in arterial combined micro-macrothrombotic syndrome. Both are combined diseases of the underlying EA-VMTD caused by endotheliopathy occurring in sepsis or other illnesses and additional vascular injury caused by in-hospital vascular accesses. However, their clinical phenotypes are very different. In the venous system of sepsis, vEA-VMTD exists as “silent” microthrombi (e.g., ITP-like syndrome) and additional vascular injury produces fibrin meshes (fibrin clots). However, physiologically “silent” microthrombi and fibrin meshes can unify together via macrothrombogenesis and produce the clinical phenotype of proximal/central DVT (i.e., VTE). In the arterial system with different circulatory hemodynamic (i.e., shear stress flow) and physiologic function (i.e., oxygen delivery), “microangiopathic” microthrombi interact fibrin meshes and unify together via macrothrombogenesis at the multiple small arteries. It produces the clinical phenotype of multiple peripheral gangrene. These characteristic differences create “peripheral gangrene” in arterial system and “VTE” in the venous system even though both are the same combined micro-macrothrombosis as delineated in Table 6. Abbreviations: CVST, cerebral venous sinus thrombosis; DVT, deep venous thrombosis; ECs, endothelial cells; EVT, extravascular tissue; ITP, immune thrombocytopenic purpura; IVCT, inferior vena cava thrombosis; MMT, micro-macrothrombosis; MODS, multiorgan dysfunction syndrome; SET, subendothelial tissue; SVCT, superior vena cava thrombosis; SVT, splanchnic venous thrombosis; TF, tissue factor; ULVWF, ultra large von Willebrand factor; vEA-VMTD venous endotheliopathy-associated vascular microthrombotic disease; VTE, venous thromboembolism.

**Table 1 life-12-00220-t001:** Clinical features of two phenotypes of venous thrombosis.

Examples of Phenotypes	Distal DVT (De Novo Venous Macrothrombosis)	Proximal/Central DVT (Combined Venous Micro-Macrothrombosis)
**Disease Examples**		
Distal DVT		
	Solitary DVT	
Complex DVT		
		Proximal/central DVT
		VTE/PTE
		CVST
		IVCT/SVCT
		Budd–Chiari syndrome
		Splanchnic vein thrombosis
**Clinical Features Example in DVT**		
Size	Small	Large and extended
Shape	Simple	Irregular
Multiplicity/character of thrombosis	Single, and isolated	Multiple, and connected
Embolic complication	Absent or unusual	Common
Prognosis	Excellent	Very unfavorable
**Mechanisms of Vascular Injury**		
Event/Underlying disease	Local trauma (rarely with surgery/vascular access)	Underlying vEA-VMTD (e.g., sepsis) + local trauma (commonly with surgery/vascular access)
Extent of vascular damage	Local ECs/SET injury	Disseminated ECs injury + local/regional ECs/SET injury

Note: CVST, cerebral venous sinus thrombosis; DVT, deep vein thrombosis; vEA-VMTD, venous endotheliopathy-associated vascular microthrombotic disease; ECs, endothelial cells; EVT, extravascular tissue; PTE, pulmonary thrombo-embolism; SET, subendothelial tissue.

**Table 2 life-12-00220-t002:** Three essentials in thrombosis and thrombogenesis.

1—Genesis of Primary Phenotypes of Thrombosis
(1) The phenotype is determined by the functional character of the injured vessel (i.e., vascular pressure and oxygen/CO_2_ carrying function: arterial or venous vasculature) (2) The phenotype is determined by the location of the involved vessel (i.e., involved organ or tissue: brain, lungs, heart, liver, nerve, muscle etc.) (3) The phenotype is determined by the size of the vasculature (i.e., microvasculature and larger vessel: capillary/arteriole, venule, artery, vein) (4) The phenotype is determined by the depth of vascular wall injury (i.e., ECs, SET, and/or EVT)
**2—Genesis of Secondary Phenotype of Thrombosis**
(1) The secondary phenotype is influenced by the extent of vascular tree involvement (e.g., localized injury vs. disseminated injury: e.g., sepsis vs. local trauma) (2) The secondary phenotype is influenced by the underlying genetic disorder (e.g., thrombophilia: e.g., ADAMTS13 deficiency, protein C deficiency, FV-Leiden) (3) The secondary phenotype is influenced by the iatrogenic vascular injury (e.g., hospitalization related vascular injury such as surgery, and vascular device/access) (4) The secondary phenotype is influenced by inappropriate treatment (e.g., platelet transfusion [e.g., for ITP], or recombinant FVIIa treatment causing fibrin clot disease)
**3—Basic and Combined Phenotypes of Thrombosis**
(1) Microthrombosis, macrothrombosis, fibrin clot disease, and hematoma ^a^ (e.g., EA-VMTD, TTP-like syndrome, MODS, AT, ITP, isolated DVT, isolated PT, APL, hemarthrosis ^a^) (2) Multiple combined micro-macrothrombosis (i.e., micro-macrothrombosis with gangrene: e.g., SPG, PDIS, ANF, acrocyanosis, Fournier’s disease; micro-macrothrombosis with VCCS: e.g., VTE, PTE, CVST) (3) Complex phenotypes with underlying VMTD and additional localized vascular injury and genetic thrombophilia (e.g., various phenotypes of “DIC” with microthrombosis and hepatic coagulopathy with/without thrombophilia, Kawasaki disease with inflammation, purpura fulminans)

Note: APL, acute promyelocytic leukemia; ANF, acute necrotizing fasciitis; AT arterial thrombosis; VCCS, venous circulatory congestion syndrome; CVST, cerebral venous sinus thrombosis; “DIC”, “false” disseminated intravascular coagulation; DVT, deep vein thrombosis; EA-VMTD, endotheliopathy-associated vascular microthrombotic diseases; ECs, endothelial cells; EVT, extravascular tissue; ITP, immune thrombocytopenic purpura; IVCT, inferior vena cava thrombosis; MODS, multiorgan dysfunction syndrome; PDIS, peripheral digit ischemic syndrome; PT, pulmonary thrombosis; PTE, pulmonary thromboembolism; SET, subendothelial tissue; SPG, symmetrical peripheral gangrene; TTP, thrombotic thrombocytopenic purpura; VTE, venous thromboembolism; ^a^ indicates extravascular blood clots.

**Table 3 life-12-00220-t003:** Two faces of venous thrombosis with pathologic, pathogenetic and laboratory differences based on two-path unifying theory of hemostasis and endothelial molecular pathogenesis (Reproduced with permission from Chang JC: *Medicina (Kaunas).* 2021 Oct 26;57(11):1163).

Phenotypes	Distal DVT (De Novo Venous Macrothrombosis)	Proximal/Central DVT (Combined Venous Micro-Macrothrombosis)
**Disease Examples**		
Venous thrombosis		
DVT	Distal DVT	
VTE		Proximal/central, multiple and/or large/long DVT
		Multiple VTE; PTE
		IVCT; SVT; PVT; BCS; SVCT; CVST
Other complex venous thrombosis		
**Mechanisms of Vascular Injury**		
Events	Local trauma (rarely with surgery/vascular access)	Underlying disease (vEA-VMTD [e.g., sepsis]) + local trauma (commonly with surgery/vascular access)
Extent of vascular damage	Local ECs/SET injury	Disseminated ECs injury + local/regional ECs/SET injury
**Pathogenesis**		
Activated thrombosis path	ULVWF and TF paths from local trauma	ULVWF path from vEA-VMTD and TF path from regional trauma
Thrombi character	Macrothrombus	Combined “microthrombi strings–fibrin meshes”
Severity	Typically, solitary and self-limited	Serious and often with multiple/large thrombi
Severe inflammation	Absent	May be present and can be severe
Venous disseminated EA-VMTD	Absent	Commonly present (e.g., ITP-like syndrome)
MOIS	Absent	Commonly present
**Diagnostic Findings/Markers**		
ITP-like syndrome	Does not occur	Sometimes occurs
ULVWF/VWF/VWF antigen	Normal	Overexpressed
FVIII activity	Normal	Increased
ADAMTS13 activity	Normal	Mild to moderately decreased
D-dimer	Normal	Markedly increased
Immune: ANA; APLA; Anti-	Negative	May be positive
dsDNA; Anti-PF4A		
**Therapeutic Approach per Theory**	No treatment or short-term anticoagulant	Anticoagulant and antimicrothrombotic agent (?)

Note: Anti-PF4A, antiplatelet factor 4 antibodies; APLA, antiphospholipid antibodies; BCS, Budd–Chiari syndrome; CVST, cerebral venous sinus thrombosis; DVT, deep vein thrombosis; vEA-VMTD, venous endotheliopathy-associated vascular microthrombotic disease; ECs, endothelial cells; EVT, extravascular tissue; ITP, immune thrombocytopenic purpura; IVCT, inferior vena cava thrombosis, MODS, multiorgan dysfunction syndrome; PTE, pulmonary thromboembolism; PVT, portal vein thrombosis; SET, subendothelial tissue; SVCT, superior vena cava thrombosis; SVT, splanchnic vein thrombosis; TF, tissue factor; ULVWF, ultra large von Willebrand factor; VTE, venous thromboembolism; VWF, von Willebrand factor.

**Table 4 life-12-00220-t004:** Clinical phenotypes and mechanisms of endotheliopathy in arterial and venous systems per “two-path unifying theory” of hemostasis (reproduced from Vaccine-Associated Thrombocytopenia and Thrombosis: Venous Endotheliopathy Leading to Venous Combined Micro-Macrothrombosis with permission from Chang JC: *Medicina (Kaunas).* 26 October 2021; 57(11):1163).

Clinical Phenotype	Arterial Endotheliopathy	Venous Endothelipathy
	aEA-VMTD	vEA-VMTD
**Underlying pathology**	Efferent circulation from the heart (oxygen delivery)	Afferent circulation into the heart and lungs (CO2 disposal)
**Physiological/hemodynamic difference**	High pressure flow	Low pressure flow
	High shear stress	Low shear stress
	Capillary and arteriolar microvascular event	Venous and pulmonary microvascular event
**Primary cause**		
Vascular injury (ECs)	Sepsis-induced microvascular endotheliopathy	Sepsis-induced venous endotheliopathy
		Vaccine-induced venous endotheliopathy
Vascular pathology site	Disseminated aEA-VMTD at microvasculature	Transient or “silent” vEA-VMTD at venous system
Activated hemostatic path	ULVWF path	ULVWF path
Thrombosis component	Microthrombi strings	Microthrombi strings
**Clinical phenotypes**	TTP-like syndrome	ITP-like syndrome
	- consumptive thrombocytopenia - MAHA - MODS/MOIS	- consumptive thrombocytopenia - MOIS

Note: aEA-VMTD, arterial endotheliopathy-associated vascular microthrombotic disease; vEA-VMTD, venous-VMTD; ECs, endothelial cells; ITP, immune thrombocytopenic purpura; MAHA, microangiopathic hemolytic anemia; MODS, multiorgan dysfunction syndrome; MOIS, multiorgan inflammatory syndrome; TTP, thrombotic thrombocytopenic purpura; ULVWF, ultra large von Willebrand factor.

**Table 5 life-12-00220-t005:** Classification of thrombosis based on its intrinsic character and different pathogenetic mechanisms.

Character Phenotypes	Microthrombosis	Macrothrombosis	Combined Micro-macrothrombosis
**Vascular Tree System**			
Arterial	aEA-VMTD (e.g., TIA, TCIP, TTP-like syndrome, MODS such as DES, HUS, ANP, RML)	Arterial macrothrombosis (e.g., AIS, aortic thrombosis)	Gangrene (arterial combined MMT) (e.g., SPG, multiple digit gangrene, limb gangrene, PF, acrocyanosis, Burger’s disease, Fournier’s disease, necrotizing fasciitis, diabetic gangrene)
Venous	vEA-VMTD (e.g., ITP, TCIP, PNH, ARDS, hepatic VOD)	Venous macrothrombosis (e.g., distal DVT, solitary DVT)	Multiple DVT (venous combined MMT) (e.g., proximal/central DVT, VTE, PTE, CVST, IVCT, SVCT, SVT, PVT, Budd-Chiari syndrome)
**Vascular Wall System**			
Involved vessel	Capillary/arteriole Venule Vasa vasorum	Small artery Vein -	Multiple small arterial vasculatures Multiple extended venous vasculatures
Thrombus character	Microthrombi strings	Macrothrombus	Microthrombi strings-fibrin meshes Combined micro-macrothrmbi
Pathogenetic factors	Endotheliopathy (ECs) (e.g., sepsis, diabetes)	Trauma (ECs +SET +/− EVT) (e.g., vascular access, surgery)	Endotheliopathy + traumatic vascular injury (e.g., sepsis + vascular access)
**Damaged Vessel**			
Depth of injury	ECs	ECs + SET +/− EVT	ECs + SET +/− EVT
Location	Generalized or localized	Localized or solitary	Multiple or regionalized
Hemostatic components	ULVWF + platelet	ULVWF+TF + fibrin	ULVWF + TF + fibrin + NETs

Involved hemostatic path	ULVWF	ULVWF and TF	ULVWF, TF with NETosis

Note: aEA-VMTD; arterial endotheliopathy-associated vascular microthrombotic disease; vEA-VMTD, venous EA-VMTD; AIS, acute ischemic stroke; ARDS, acute respiratory distress syndrome; CVST, cerebral venous sinus thrombosis; DVT, deep vein thrombosis; ECs, endothelial cells; EVT, extravascular tissue; ITP, idiopathic (immune) thrombocytopenic purpura; IVCT, inferior vena cava thrombosis; MMT, micro-macrothrombosis; NETs, neutrophil extracellular traps; PVT, portal vein thrombosis; SET, subendothelial tissue; SPG, symmetrical peripheral gangrene; SVCT, superior vena cava thrombosis; SVT, splanchnic vein thrombosis; TCIP, thrombocytopenic in critically ill patients; TF, tissue factor; TIA, transient ischemic attack; VTE, venous thromboembolism; ULVWF, ultra large von Willebrand factor; VOD, veno-occlusive disease.

**Table 6 life-12-00220-t006:** Pathogenetic features of combined micro-macrothrombosis in arterial and venous systems per “two-path unifying theory” of hemostasis.

Clinical Phenotype	Arterial Combined Micro-Macrothrombosis	Venous Combined Micro-Macrothrombosis
**Pathologic nature of thrombosis**		
(Examples)	Gangrene:	Venous circulatory congestion syndrome:
	SPG, limb gangrene, Fournier’s disease, PF, diabetic gangrene, necrotizing fasciitis	Proximal/central DVT, VTE, PTE, CVST, PVT, IVCT/SVCT, SVT, Budd-Chiari syndrome
**Primary event**		
Vascular injury (ECs)	Diseases causing microvascular endotheliopathy (e.g., sepsis, diabetes, pregnancy)	Diseases causing venous endotheliopathy (e.g., sepsis, diabetes, pregnancy, autoimmunity)
Vascular pathology site	Disseminated aEA-VMTD at terminal arterial tree	Regional vEA-VMTD at vascular injury site
Activated hemostatic path	ULVWF path	ULVWF path
Thrombosis component	Microthrombi strings	Microthrombi strings
**Secondary event**		
Vascular injury (SET)	Arterial vascular damage (e.g., surgery, vascular accesses/devices)	Venous vascular damage (e.g., surgery, vascular accesses/devices)
Pathology	Fibrin clots in arterial system	Fibrin clots in venous system
Vascular pathology site	Terminal arterial vasculature tree	Large vein or pulmonary artery
Activated hemostatic path	TF path	TF path
Thrombosis component	Fibrin meshes	Fibrin meshes
**Pathogenesis**		
Mechanism	Unifying of microthrombi string and fibrin meshes in arterial system as minute macrothrombi	Unifying of microthrombi strings and fibrin meshes in venous system often as connected macrothrombi
Thrombosis character	Microthrombi strings-fibrin meshes complex	Microthrombi strings-fibrin meshes complex
Thrombosis form and effect	Multiple terminal efferent digit anoxia and peripheral gangrene	Connected and regional, with circulatory afferent flow with venous circulatory congestion syndrome

Note: aEA-VMTD, arterial endotheliopathy-associated vascular microthrombotic disease; vEA-VMTD, venous-VMTD; CVST, cerebral venous sinus thrombosis; ECs, endothelial cells, DVT, deep venous thrombosis; IVCT, inferior vena cava thrombosis; PF, purpura fulminans; PVT, portal vein thrombosis; SET, subendothelial issue; SPG, symmetrical peripheral gangrene; SVCT, superior vena cava thrombosis; SVT, splanchnic vein thrombosis; TF, tissue factor; PTE, pulmonary thromboembolism; ULVWF, ultra large von Willebrand factor: VTE, venous thromboembolism.

**Table 7 life-12-00220-t007:** Novel classification of venous thrombosis based on in vivo hemostatic theories.

	Benign DVT		Serious DVT		
Phenotype	Distal DVT	“Silent” vEA-VMTD	VTE	PTE	“Serious” vEA-VMTD
Example	Calf DVT (local)	ITP-like syndrome	PVT; SVC;/IVCT; SVT	PTE from VTE	Hepatic VOD
			CVST		PNH
**Clinical characteristics**					
Cause	Trauma	Septic endotheliopathy	Endotheliopathy + vascular access	Endotheliopathy + vascular access	Endotheliopathy
Genesis	Local vascular injury	vEA-VMTD	vEA-VMTD + injury	vEA-VMTD +injury	Transplant (VOD) PIGA gene mutation (PNH)
**Thrombosis character**	Macrothrombus	Microthrombi strings	Combined micro-macrothrombi	Combined micro-macrothrombi	Microthrombi strings
**Activated hemostatic path**	ULVWF + TF	ULVWF	ULVWF + TF	ULVWF + TF	ULVWF
**Laboratory characteristics**					
Thrombocytopenia	Absent	Common	Common	May be present	Always present
VWF/FVIII activity	Normal	Mildly increased	Increased	May be increased	Markedly increased
ANA	Negative	May be positive	May be positive	May be positive	Unknown
Anti-PLA/Anti-PF4A	Negative	May be positive	May be positive	May be positive	Unknown
ADAMTS13 activity	Normal	May be decreased	Decreased	May be decreased	Decreased (?)
**Proposed treatment**	None/short-term anticoagulation	No treatment or IVIG	Anticoagulant + antimicrothrombotic agent	Anticoagulant + antimicrothrombotic agent	Antimicrothrombotic Agent (?)

Note: ANA, antinuclear antibody; CVST, cerebral venous sinus thrombosis; DVT deep vein thrombosis; ITP, immune thrombotic thrombocytopenia; IVCT, inferior vena cava thrombosis; PF4A; platelet factor 4 antibodies; PLA, phospholipid antibodies; PNH, paroxysmal nocturnal hemoglobinuria; PT, pulmonary thrombosis; PTE, pulmonary thromboembolism; PVT, portal vein thrombosis; SVCT, superior vena cava thrombosis; SVT, splanchnic vein thrombosis; TF, tissue factor; ULVWF, unusually large von Willebrand factor; vEA-VMTD, venous endotheliopathy-associated vascular microthrombotic disease; VOD, veno-occlusive disease; VWF, von Willebrand factor multimers.

## Data Availability

Data sharing not applicable to this article as no datasets were generated or analyzed during the current study.

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
