# Peer review of "Pathogenesis of Two Faces of DVT: New Identity of Venous Thromboembolism as Combined Micro-Macrothrombosis via Unifying Mechanism Based on “Two-Path Unifying Theory” of Hemostasis and “Two-Activation Theory of the Endothelium”"

_life, 2022, doi:10.3390/life12020220_

Round 1

Reviewer 1 Report

Interesting manuscript, well supported with clear discussion. I appreciated the synthetic style of work development that helps to remind presented data. The paper is well written and formally correct and it is clear its clinical relevance, and what this article should add to the body of knowledge on this topic

Author Response

Comments and Suggestions for Author

Interesting manuscript, well supported with clear discussion. I appreciated the synthetic style of work development that helps to remind presented data. The paper is well written and formally correct and it is clear its clinical relevance, and what this article should add to the body of knowledge on this topic

Author’s response

Thank you very much for your comments. Indeed, DVT is an urgent issue in medicine and is characterized by two different pathogeneses, which has profound implication in the management. I identified VTE from novel hemostatic mechanism of “two-path unifying theory”.

Reviewer 2 Report

In this manuscript, the author does an effort to define two faces of Deep Venous Thrombosis. The manuscript uses many definitions and concepts, most of which are ill-defined.

It is positive that the author does an effort to shed new light on thrombosis, but my main 3 concerns are the following:

  • He should use better-defined definitions and concepts and use them consistently throughout the text
  • He should reduce the text overall by about 50%, bringing it back to 14-15 pages, ref included
  • It should be read by a native English speaker, which apparently did not happen, which is surprising since the author is situated in California.

Points of concern:

  1. The author refers to injury as a cause of venous thrombosis. He does not mention stasis or hypoxia, which he dismisses. There is hardly any “physiological” cause of thrombosis, thrombosis being the result of disturbed haemostasis. The distinctions the author makes throughout his plea are for “pathological” thrombosis. Then, he should involve other causes for dysregulated haemostasis as mechanical injury only.
  2. It seems logical to define “microthrombosis" as thrombosis occurring in small blood vessels, as opposed to macrothrombosis. But then he should define well where this happens and be consistent throughout. In some place, thrombi are described as large or small, depending on their localization. This would be correct, if also correctly coupled to the definition of microthrombosis throughout. This I found to be confusing in several places.
  3. The concept of Ultralarge VWF is also used in a non-conventional manner. The release of ULVWF molecules from Weibel-Palade bodies is coupled to inflammatory conditions, favoring the release of large VWF multimers, having a more pronounced effect on platelets. This happens primarily in conditions where shear forces help to unfold these multimers (temporarily sticking to the endothelial cells from which they are released). This implies that all this is predominant in small arteries and arterioles, but less in venules. The author needs to clarify the differences and similarities with the venous circulation, where inflammation plays a role and where ADAMTS13 can be active, indeed. But VWF plays a smaller role in the venules than in arterioles.
  4. The author mentions endothelial injury, but does not discuss procoagulant triggers vs the existence of an anticoagulant system in the endothelium as a source of coagulation dysfunction.
  5. Large definitions as “Two-Path Unifying Theory” and “Two-Activation Theory of the Endothelium” seem to be excessive in a title. Again, good definitions are needed, based on evidence, then justifying new insights.
  6. TTP is typically a disease of the superficial arteriolar circulation. When the author wants to draw parallels with venous conditions, he will have to convincingly draw up the mechanistic reasons to justify this.
  7. New definitions as aEA-VMTDdo not contribute to understanding. There are too many of these.

So, the authors should do an effort to present his view in a simplified manner, with less definitions, but precise and well-defined definitions, relying on established concepts. When he wants to challenge these, he will have to provide convincing evidence, justifying his plea.

Author Response

Response to Reviewer 2

Comments and Suggestions for Author

In this manuscript, the author does an effort to define two faces of Deep Venous Thrombosis. The manuscript uses many definitions and concepts, most of which are ill-defined.

It is positive that the author does an effort to shed new light on thrombosis, but my main 3 concerns are the following:

  • He should use better-defined definitions and concepts and use them consistently throughout the text
  • He should reduce the text overall by about 50%, bringing it back to 14-15 pages, ref included
  • It should be read by a native English speaker, which apparently did not happen, which is surprising since the author is situated in California.

Author’s response

It is true that I have used several novel concepts such as 1) “two-path-unifying theory of hemostasis, 2) molecular pathogenesis of endotheliopathy, 3) vascular injury as a sole trigger event in thrombogenesis, 4) unifying mechanism of microthrombosis and macrothrombosis, 5) interaction of vascular wall damage on the thrombogenesis, and others. All of these concepts are clearly defined in this text and my previous publications even to the molecular levels. That is the main reason why this Review is long and complex.

Reviewer’s Points of concern

  1. The author refers to injury as a cause of venous thrombosis. He does not mention stasis or hypoxia, which he dismisses. There is hardly any “physiological” cause of thrombosis, thrombosis being the result of disturbed haemostasis. The distinctions the author makes throughout his plea are for “pathological” thrombosis. Then, he should involve other causes for dysregulated haemostasis as mechanical injury only.

Author’s response

I have clearly defined the mechanisms of “thrombosis” in the text. Now, it is clear that Virchow’s triad is based on “incorrect” concept because thrombosis never occurs without the release of ULVWF and TF from ECs and SET damage of the vessel wall. Stasis is not the risk factor for “thrombosis” but is only risk factor for “vascular injury”. Hypoxia is the result of “thrombosis”, but not the cause of “thrombosis”.

  1. It seems logical to define “microthrombosis" as thrombosis occurring in small blood vessels, as opposed to macrothrombosis. But then he should define well where this happens and be consistent throughout. In some place, thrombi are described as large or small, depending on their localization. This would be correct, if also correctly coupled to the definition of microthrombosis throughout. This I found to be confusing in several places.

Author’s response

The difference between “microthrombosis” and “macrothrombosis” is nothing to do with thrombosis located in the different size of the blood vessel. I have succinctly defined as “microthrombi are composed of platelets-ULVWF complexes by lone activation of ULVWF path of hemostasis due to endotheliopathy (e.g., sepsis). They are microthrombi strings and always “disseminated” in the capillaries and arterioles as seen in sepsis (Figure 1), but macrothrombi do not occur as disseminated disease, but as local thrombosis at vascular injury site (e.g., distal DVT). Macrothrombi can also be multiple “minute” macrothrombi composed of “local” microthrombi-fibrin clot complex in gangrene syndrome (e.g., SPG or VTE) or large macrothrombus (e.g., acute ischemic stroke).

  1. The concept of Ultralarge VWF is also used in a non-conventional manner. The release of ULVWF molecules from Weibel-Palade bodies is coupled to inflammatory conditions, favoring the release of large VWF multimers, having a more pronounced effect on platelets. This happens primarily in conditions where shear forces help to unfold these multimers (temporarily sticking to the endothelial cells from which they are released). This implies that all this is predominant in small arteries and arterioles, but less in venules. The author needs to clarify the differences and similarities with the venous circulation, where inflammation plays a role and where ADAMTS13 can be active, indeed. But VWF plays a smaller role in the venules than in arterioles.

Author’s response

ULVWF is exactly the same in their role of hemostasis and thrombosis. Until now, it has not been known that “ULVWF” interacting with platelets forms “platelet-ULVWF complexes (microthrombi strings). Low molecular weight VWF is not the one causing hemostasis and thrombosis but ULVWF multimers are. They do not need shear stress force in traumatic vascular injury. Natural ULVWF which is highly adhesive to platelets and released from damaged blood cells attracts platelets to start microthrombogenesis. Likewise, ULVWF released from septic endotheliopathy also attracts platelets in septic endotheliopathy. Because of their microthrombotic nature, it happens in the capillaries and arterioles. Inflammatory pathway is independent from microthrombogenesis in endotheliopathy (Figure 2). The effects of venous circulation and arterial circulation is clearly explained in Table 4.

  1. The author mentions endothelial injury, but does not discuss procoagulant triggers vs the existence of an anticoagulant system in the endothelium as a source of coagulation dysfunction.

Author’s response

Endothelial injury triggers the release of ULVWF multimers, which is the essential component activating ULVWF (microthrombotic) path of hemostasis (Figure 1 and Figure 2). There are no such systems as “procoagulant system” or “anticoagulant system” in hemostasis, but alteration of participating components in hemostasis changes coagulation and thrombogenesis by gene mutation (e.g., ADAMTS13 deficiency, protein C-deficiency) and certain disease state (e.g., loss of protein S by nephrotic syndrome and release of ULVWF endotheliopathy) and produces abnormal thrombosis as shown in Table 2.

  1. Large definitions as “Two-Path Unifying Theory” and “Two-Activation Theory of the Endothelium” seem to be excessive in a title. Again, good definitions are needed, based on evidence, then justifying new insights.

Author’s response

One will not be able to comprehend normal hemostasis and abnormal hemostasis in the bleeding disorder and the mechanism of vascular injury producing thrombogenesis without the understanding of “Two-Path Unifying Theory” and “Two-Activation Theory of the Endothelium”. In my previous publications, I have explained in great detail how I have derived these two concepts in in vivo vascular models rather than test tube models or cell-based models. These theories solved so many undetermined hematologic disorders such as TTP-like syndrome, endotheliopathy, ARDS, sepsis, “DIC”, “ITP”, thrombo-hemorrhagic syndrome, different phenotypes of stroke, diabetic ketoacidosis, PNH, complicated coagulopathy and others as well as combined micro-macrothrombotic syndromes.

  1. TTP is typically a disease of the superficial arteriolar circulation. When the author wants to draw parallels with venous conditions, he will have to convincingly draw up the mechanistic reasons to justify this.

Author’s response

VMTD is characterized by “microthrombosis”. However, the concept of VMTD may be difficult one to understand for some clinicians: first, the pathogenesis of this disorder can be caused by 1) ADAMTS13 deficiency (i.e., hereditary TTP and autoimmune disease due to ADAMTS13 antibodies) and 2) endotheliopatrhy (i.e., TTP-like syndrome in sepsis and other critical illnesses). Second, the expression of VMTD is influenced by the hemodynamic difference of the vascular system: 1) arterial system producing the triad of thrombocytopenia, MAHA, and MODS and 2) venous system producing thrombocytopenia only (“silent ITP). I have painfully explained the 2nd concept of difference between arterial and venous systems in several ways in my manuscript. Therefore, simple term “microthrombosis” does not represent a disease. It can be TTP, TTP-like syndrome, and ITP, which concepts are also important for the understanding of combined “micro-macrothrombosis and their syndromes).

  1. New definitions as aEA-VMTDdo not contribute to understanding. There are too many of these. So, the authors should do an effort to present his view in a simplified manner, with less definitions, but precise and well-defined definitions, relying on established concepts. When he wants to challenge these, he will have to provide convincing evidence, justifying his plea.

Author’s response

The response to this comment from Reviewer 3 is already answered the above (6) Response. Without the terms “VMTD”, “EA-VMTD”, aEA-VMTD”, and “vEA-VMTD”, clinicians will not be able to understand the true pathology and clinical phenotypes of generic term “microthrombosis” as well as combined thrombotic syndromes that I have explained in Table 2. I do not justify with plea. I have correlated each concept derived from clinical findings, and laboratory data, and published experimental data and proved with my hemostat5ic theories.

I have appreciated this opportunity to answer to your comments because your points can be addressed in my next articles. Thanks for your helpful queries.

Reviewer 3 Report

It'a an interesting and thorough review.

Author Response

Response to Reviewer 3

Comments and Suggestions for Authors

It's an interesting and thorough review.

Author’s response

Thank you for your kind words.  This new paradigm on conceptual differences in two DVT will have significant implication for prevention and treatment for so many patients suffering from DVT. Thanks to you.

Reviewer 4 Report

The author of the current manuscript has made a thorough review of the possible pathogenetic mechanisms of deep venous thrombosis, using data from published COVID-19  literature.

I have the following recommendations, that I believe will improve the scientific quality of the article:

The title of the manuscript is too long – please, shorten it to become more “attractive”.

Please, make the abstract more succinct.

Line 57-58: “…involving the veins of the circulatory system.” – it does not sound well. Please, modify it  (for instance “involving venous vessels” or similar).

Line 58: “Following an intravenous vascular injury venous thrombosis can develop…” – actually, change of any component of the so called “Virchow’s triad” can cause venous thrombosis. Please, specify/modify this part of the text.

Citations like “[4,6,7,8,9,10]” should be presented this way “[ 4, 6-10]”. Please, correct them.

Line 95: “The mystery of various thrombotic syndromes…” – it does not sound very well and should be modified, for example “The unknown (or unspecified/unclear, etc.) pathogenesis (or nature) of  thrombotic syndromes in patients, infected with COVID-19….”. The same for “…has been a great challenge to this author and led me to reexamine…”. It could be “…gave us grounds to investigate/explore…”  or something similar”

In the footer of the tables, the author can omit “Abbreviations:” – it is clear what it is.

Generally, the entire manuscript is too extensive  - it shoud be shortened to be more readable and attractive.

Author Response

Response to Reviewer 4

The author of the current manuscript has made a thorough review of the possible pathogenetic mechanisms of deep venous thrombosis, using data from published COVID-19 literature.

I have the following recommendations, that I believe will improve the scientific quality of the article.

I have the following recommendations, that I believe will improve the scientific quality of the article:

Comments and Suggestions  and Author’s response

  1. The title of the manuscript is too long – please, shorten it to become more “attractive”.Please, make the abstract more succinct.

Author’s response: I have shortened the title for easy reading.

  1. Line 57-58: “…involving the veins of the circulatory system.” – it does not sound well. Please, modify it (for instance “involving venous vessels” or similar).

Author’s response: I have changed to “venous vessels”

  1. Line 58: “Following an intravenous vascular injury venous thrombosis can develop…” – actually, change of any component of the so called “Virchow’s triad” can cause venous thrombosis. Please, specify/modify this part of the text.

Author’s response:  The truth is “Virchow’s triad” is incorrect, which I have discussed many times in my previous publications. The “vascular stasis” due to a long airplane trip or “hemostatic alteration” such as thrombophilia does not cause intravascular “thrombosis” without a vascular injury that releases ULVWF multimers and/or TF. However, they are contributing factors for “vascular injury” and also accelerate “thrombosis” formation once ULVWF and TF are released from the vessel wall. Therefore, the only risk factor for thrombosis is “vessel injury”, but “blood stasis”, “thrombophilia”, “trauma”, “hospitalization” and “old age”, etc. are the risk factors for “vessel injury”.  This is the most important concept in the mechanism of “thrombogenesis and thrombosis”. I have discussed it in a great length in my other papers and here too.

  1. Citations like “[4,6,7,8,9,10]” should be presented this way “[ 4, 6-10]”. Please, correct them.

Author’s response: I have used this method purposely to check references carefully, but now changed to the format you have suggested.

  1. Line 95: “The mystery of various thrombotic syndromes…” – it does not sound very well and should be modified, for example “The unknown (or unspecified/unclear, etc.) pathogenesis (or nature) of thrombotic syndromes in patients, infected with COVID-19….”. The same for “…has been a great challenge to this author and led me to reexamine…”. It could be “…gave us grounds to investigate/explore…”  or something similar”

Author’s response: I have modified the sentence as you have suggested.

  1. In the footer of the tables, the author can omit “Abbreviations:” – it is clear what it is.

Author’s response: I am well aware that this practice is atypical one, but I do feel this “Abbreviations” at the footer of Tables and Figures definitely helps the readership when the article is long and comprehensive with the use of so many abbreviations involving many different fields of medical specialty and subspecialty. This practice enhances the understanding of complexity of the article and prevents the confusion of the readership. I have used it in many journals, but neither editors nor publishes have objected.

Many thanks for your thoughtful suggestions. I am very grateful to your intertest in my article.

Round 2

Reviewer 2 Report

None

Reviewer 4 Report

The authors have observed the recommendations of the reviewers and have significantly improved their manuscript. It is now ready to be published in its current version.